# Transcriptomic Immune Profiles Can Represent the Tumor Immune Microenvironment Related to the Tumor Budding Histology in Uterine Cervical Cancer

**DOI:** 10.3390/genes13081405

**Published:** 2022-08-07

**Authors:** Tan Minh Le, Hong Duc Thi Nguyen, Eunmi Lee, Donghyeon Lee, Ye Seul Choi, Junghwan Cho, Nora Jee-Young Park, Hyung Soo Han, Gun Oh Chong

**Affiliations:** 1Department of Biomedical Science, Graduate School, Kyungpook National University, Daegu 41944, Korea; 2BK21 Four Program, School of Medicine, Kyungpook National University, Daegu 41944, Korea; 3Clinical Omics Institute, Kyungpook National University, Daegu 41405, Korea; 4Department of Pathology, Kyungpook National University, Chilgok Hospital, Daegu 41404, Korea; 5Department of Physiology, School of Medicine, Kyungpook National University, Daegu 41944, Korea; 6Department of Obstetrics and Gynecology, Kyungpook National University, Chilgok Hospital, Daegu 41404, Korea

**Keywords:** cervical cancer, gene expression, immune-related genes, tumor budding, tumor immune microenvironment

## Abstract

Tumor budding (TB) histology has become a critical biomarker for several solid cancers. Despite the accumulating evidence for the association of TB histology with poor prognosis, the biological characteristics of TB are little known about in the context related to the tumor immune microenvironment (TIME) in uterine cervical cancer (CC). Therefore, this study aimed to identify the transcriptomic immune profiles related to TB status and further provide robust medical evidence for clinical application. In our study, total RNA was extracted and sequenced from 21 CC tissue specimens. As such, 1494 differentially expressed genes (DEGs) between the high- and low-TB groups were identified by DESeq2. After intersecting the list of DEGs and public immune genes, we selected 106 immune-related DEGs. Then, hub genes were obtained using Least Absolute Shrinkage and Selection Operator regression. Finally, the correlation between the hub genes and immune cell types was analyzed and four candidate genes were identified (one upregulated (*FCGR3B*) and three downregulated (*ROBO2*, *OPRL1*, and *NR4A2*) genes). These gene expression levels were highly accurate in predicting TB status (area under the curve >80%). Interestingly, *FCGR3B* is a hub gene of several innate immune pathways; its expression significantly differed in the overall survival analysis (*p =* 0.0016). In conclusion, *FCGR3B*, *ROBO2*, *OPRL1*, and *NR4A2* expression can strongly interfere with TB growth and replace TB to stratify CC patients.

## 1. Introduction

Uterine cervical cancer (CC) is one of the most common cancers in women [1]. Although progress has been made in CC prevention and management, it still has a poor outcome. According to Globocan 2020, 3218 new CC cases and 1014 CC deaths occur annually in Korea [1]. Therefore, novel markers for CC diagnosis, prognosis, and treatment have long been of great interest.

Recently, tumor budding (TB) histology has become a critical biomarker for several solid cancers, including colorectal cancer, head and neck cancer, pancreatic cancer, etc. [2,3,4,5,6]. In addition, we previously showed that TB status is a potential independent prognostic factor and can strongly interfere with CC treatment strategies [7]. However, the main disadvantage of this histopathological marker is that it can only be established after surgery. In this scenario, insights into the biological characteristics of TB can provide a helpful direction to investigate potential tools for CC management.

The tumor microenvironment (TME), referred to as a “house” of cancer cells, comprises non-resident components (tumor-derived cells and infiltrating leukocytes) and resident components (blood vessels, nerve fibers, mesenchyma, and structural components) [8,9,10]. Some researchers suggested six TME subgroups, including hypoxic niche, immune microenvironment, metabolism microenvironment, acidic niche, innervated niche, and mechanical microenvironment [8]. The tumor immune microenvironment (TIME) is an important aspect because the theory of TIME can provide a powerful account of how to develop effective anti-tumor therapies. It is revealed that the immune response in TME can sometimes block cancer development [11]. Unfortunately, in many cases, tumor cells often activate the immunosuppressive mechanisms and lead to tumor escape from the host immune response [8,9,10].

Until now, TB has been little known in relation to the TIME. Nearchou (2019) and Dawson (2020) found that CD8+/CD3+ T-cell location and density are correlated with TB status in colorectal cancer [12,13]. However, there have not been any published studies about the relationship between tumor buds and TIME in CC. Therefore, this study aimed to identify the transcriptomic immune profiles related to TB status and further provide robust medical evidence for clinical application, such as stratification, prognosis, and immune-targeted therapeutic strategy.

## 2. Materials and Methods

### 2.1. Sample Collection

This study was conducted between 2011 and 2018. Twenty-one tissues from early stage and locally advanced CC patients were obtained after radical hysterectomy. The exclusion criteria included a history of preoperative chemotherapy, radiotherapy, and synchronous malignancies. The 2009 International Federation of Gynecologic Obstetrics (FIGO) staging for carcinoma of the cervix was used to stage the patients [14]. The flowchart in Figure 1 represents the overall process of this study.

### 2.2. Pathological Process

Specimens were selected from the tumor area and stained with hematoxylin and eosin. All tissue slices (ranging from 8 to 25 per patient) were carefully assessed for histopathological features. TB is described as a single neoplastic cell or cell cluster of up to four neoplastic cells at the invasive front of the tumor [15]. High TB was defined as ≥5 buds/high-power field (Appendix A). Detailed examination methods and high-TB criteria have been described previously [7,16].

### 2.3. Clinical Parameters and Follow-Up

Clinicopathological parameters included age, tumor stage (early stage < IIb and late stage ≥ IIb), histological subtype, TB status (high and low), overall survival (OS), and relapse-free survival (RFS). All patients were followed-up every three months for the first two years, every six months for the next five years, and then annually [7,16].

### 2.4. RNA Extraction and Sequencing

For each patient, total RNA was extracted from 2 to 4 sections (5 µm each) of a block using the ReliaPrep^TM^ FFPE RNA Miniprep System (Promega, Madison, WI, USA), according to the manufacturer’s protocol. Next, we conducted library preparation using the TruSeq RNA Exome Kit and RNA sequencing with NovaSeq 6000 (Illumina, San Diego, CA, USA). The quantity and quality were checked using a Qubit 4 Fluorometer (Thermo Fisher Scientific, Waltham, MA, USA), and Bioanalyzer (Aligent Technologies, Santa Clara, CA, USA).

### 2.5. Bioinformatic Analysis

Raw sequences in the FASTQ format from the sequencer were assessed for read quality using FASTQC (https://www.bioinformatics.babraham.ac.uk/projects/fastqc/, accessed on 8 September 2021). All low-quality reads and sequencing adapters were removed using Trimmomatic [17,18]. Kallisto was used to quantify RNA-seq data [19]. DESeq2 package was used to compare gene expression between high- and low-TB groups [20]. The *p* value and |log2-fold-change| thresholds were 0.05 and 1.0, respectively. A volcano plot of differentially expressed genes (DEGs) was created using the “ggplot2” package [21].

### 2.6. Immune-Related Gene Dataset

A list of 2483 public immune genes was downloaded from the ImmPort database (https://www.immport.org/shared/home/, accessed on 20 January 2022) [22]. The overlapping genes between DEGs and 2483 public immune-related genes were immune-related DEGs (IR-DEGs).

### 2.7. Functional Analysis

The IR-DEGs were uploaded to the DAVID website (https://david.ncifcrf.gov/tools.jsp, accessed on 19 April 2022) for analyzing the Gene Ontology (GO) and Kyoto Encyclopedia of Genes and Genomes (KEGG) pathways [23]. Benjamini-and-Hochberg-adjusted *p* < 0.05 was considered statistically significant.

### 2.8. Immune-Cell-Type Analysis

Gene-length-normalized expression was imported into the TIMER2.0 website (http://timer.cistrome.org/, accessed on 21 April 2022) to estimate the proportions of different immune cell types using the TIMER (TIMER immune cells) and CIBERSORT (CIBERSORT immune cells) algorithms [24,25,26]. The fraction of immune cells between the high- and low-TB groups was compared using the Wilcoxon rank-sum test and a boxplot was drawn with the “ggpubr” package [27]. Benjamini-and-Hochberg-adjusted *p* < 0.05 was considered statistically significant.

### 2.9. Candidate Gene Analysis

We performed Least Absolute Shrinkage and Selection Operator (LASSO) regression, a machine learning method in the “glmnet” package, to select hub genes from IR-DEGs [28]. Then, the correlation analysis (Spearman’s method in the “corrplot” package) between the hub genes and immune cell types was used to select the significantly correlated genes [29]. The candidate genes were the overlapping genes between CIBERSORT and TIMER. The *p* value threshold was 0.05.

### 2.10. Protein–Protein Interaction Network Analysis

To thoroughly examine the candidate gene functions, 106 IR-DEGs were uploaded to the STRING website (version 11.5) to analyze the protein–protein interaction (PPI) (https://string-db.org/, accessed on 6 May 2022) [30]. The network was visualized using Cytoscape software v3.9.1 (Shannon, P et al., Institute for Systems Biology, Seattle, WA, USA) [31].

### 2.11. Receiver Operating Characteristic (ROC) Curve and Survival Analysis

The “pROC” package was used to determine the diagnostic accuracy of the candidate genes between the high- and low-TB groups in CC [32]. Next, the optimal cut-off point for survival analysis was determined using maxstat (maximally selected rank statistics) in the “survminer“ package [33]. The different expression levels of candidate genes in OS and RFS were explored using the log-rank test in the “survival” package [34,35].

## 3. Results

### 3.1. Clinical Information

Twenty-one participants were enrolled. The clinicopathological information is shown in Table 1. The median age was 48 years (32–71 years). The early stage was dominant (81%). Fifteen and six patients (71.4% and 28.6%, respectively) belonged to the high- and low-TB groups, respectively. In addition, the distribution of pathological diagnoses was as follows: squamous cell carcinoma: 15 (71.4%); adenocarcinoma: 6 (23.8%); and adenosquamous cell carcinoma: 1 (4.8%).

### 3.2. Identification of IR-DEGs

The volcano plot indicates 1464 DEGs (929 upregulated and 535 downregulated genes) between the high- and low-TB groups (Figure 2 and Appendix A). Subsequently, 106 IR-DEGs that overlapped between the DEGs and immune-related gene database (Figure 3A) were selected. A heatmap was then used to visualize the hierarchical clustering of the identified IR-DEGs (Figure 3B and Appendix A).

### 3.3. Functional Enrichment Analysis

The functions of 106 IR-DEGs were investigated by GO enrichment and KEGG pathway analyses. As shown in Figure 4A and Appendix A, the 50 most significantly enriched genes were involved in immune response (GO:0006955, *p* = 3.2 × 10^−16^), cytokine-mediated signaling pathway (GO:0019221, *p* = 9 × 10^−13^), Fc-γ receptor signaling pathway involved in phagocytosis (GO:0038096, *p* = 5.3 × 10^−7^), external side of plasma membrane (GO:0009897, *p* = 8.2 × 10^−13^), and extracellular space (GO:0005615, *p* = 9.2 × 10^−11^). Regarding the KEGG pathway, natural-killer-cell-mediated cytotoxicity (hsa04650, *p* = 1.47 × 10^−6^), cytokine–cytokine receptor interaction (hsa04060, *p* = 1.54 × 10^−6^), and antigen processing and presentation (hsa04612, *p* = 9.5 × 10^−6^) were the main pathways (Figure 4B and Appendix A). 

### 3.4. Identification of Candidate Immune-Related Genes

#### 3.4.1. Selection of Hub Immune-Related Genes with LASSO Regression

LASSO regression was implemented for the 106 IR-DEGs. In total, 16 hub genes with the best lambda value, including *UNC93B1*, *ROBO2*, *RARB*, *PTH1R*, *PSMD14*, *OPRL1*, *NR4A2*, *LTBP3*, *LILRB3*, *KLRC2*, *IGLV4-60*, *IGKV3-7*, *IGHV3-35*, *FCGR3B*, *CSRP1*, and *AKT3*, were selected (Figure 5).

#### 3.4.2. Correlation Analysis

We first investigated the proportion of immune cells using the CIBERSORT and TIMER algorithms (Appendix A). Figure 6A (CIBERSORT) and 6B (TIMER) indicate that each immune cell type was not significantly different between the low- and high-TB groups.

Spearman’s method was used to identify the correlation between hub genes and immune cell types generated by CIBERSORT and TIMER (Figure 7). Eleven (Figure 7A) and five genes (Figure 7B) were significantly correlated with CIBERSORT and TIMER immune cells, respectively. Finally, the Venn diagram (Figure 7C) identified four candidate genes (overlapping genes), including one upregulated (*FCGR3B*) and three downregulated (*ROBO2*, *OPRL1*, and *NR4A2*) genes.

### 3.5. PPI Network Analysis

A network of 106 IR-DEGs was constructed using STRING and displayed in Cytoscape (Figure 8). The confidence score was 0.4 and the PPI enrichment *p* value was 1 × 10^−16^. *FCGR3B*, *ROBO2*, *OPRL1*, and *NR4A2* were significantly enriched in seven KEGG pathways and 51 GO terms (Table 2, Table 3, Appendix A). Interestingly, *FCGR3B* was the hub gene for the natural-killer-cell-mediated cytotoxicity and phagosome pathways.

### 3.6. ROC Curve and Survival Analysis

ROC analysis was conducted to investigate the accuracy of the four candidate genes as diagnostic biomarkers for low- and high-TB levels. The area under the curve (AUC) of *FCGR3B*, *ROBO2*, *OPRL1*, and *NR4A2* was 81.7%, 92.2%, 82.2%, and 94.4%, respectively (Figure 9). However, only the *FCGR3B* expression level significantly differed in OS (*p* = 0.0016; Figure 10).

## 4. Discussion

Recent studies have noted the high correlation between TB status and cancer progression [5,6,7,16]. Because tumor progression results from the interaction between tumor cells and the TME, insights into biological TB can provide potential tools for cancer management [5]. However, the relationship between tumor bud and TME, particularly TIME, has been unclear. Therefore, this is one of the first studies identifying the transcriptomic immune profiles of TB in CC.

In this study, 1464 DEGs were obtained by comparing the gene expression between high- and low-TB groups. By intersecting these DEGs and 2483 public immune-related genes, we selected 106 IR-DEGs. Functional analysis showed that these IR-DEGs mainly involve innate immune pathways, such as the natural killer (NK)-cell-mediated cytotoxicity, cytokine–cytokine receptor interaction, chemokine signaling, JAK-STAT, and Toll-like receptor signaling pathways. Most current reports consider CD8+T cells, FOXP3+T cells, and CD68+ macrophages to be the primary innate immune response in the tumor microenvironment [5,36]. Nevertheless, NK also plays a crucial role in the TIME. For example, Garcia-Iglesias et al. reported that NK-activating receptors and the cytotoxic activity of NK cells significantly decrease in high-grade squamous intraepithelial lesions and CC [37].

To investigate the candidate genes from the 106 IR-DEGs, multiple analyses were conducted. Briefly, the LASSO regression algorithm was implemented and 16 hub genes were identified. Then, the correlation between hub genes and immune cell types was analyzed to select candidate genes. Finally, one upregulated (*FCGR3B*) and three downregulated (*ROBO2*, *OPRL1,* and *NR4A2*) genes were identified. PPI network analysis revealed the functional insights of the four candidate genes in TB status.

First, high *FCGR3B* expression induces TB progression by suppressing NK-cell-mediated cytotoxicity and phagocytosis. *FCGR3B* encodes a low-affinity immunoglobulin γ Fc region receptor III-B protein (FcγRIIIb), a member of the IgG Fc receptor family (FcγRs) [38]. FcγRs play crucial roles in cancer immunotherapy via antibody-dependent cellular cytotoxicity and phagocytosis [38].

Previous studies have reported six human FcγRs (FcγRI/CD64, FcγRIIa/CD32a, FcγRIIb/CD32b, FcγRIIc/CD32c, FcγRIIIa/CD16a, and FcγRIIIb/CD16b) [39,40]. *FcγRIIIb/CD16b* is mainly expressed by neutrophils and also negatively regulates neutrophil activities in the TIME [39,41]. Even though the interaction between neutrophils and NK cells in TIME is unclear, neutrophils can enhance NK-derived interferon (IFN)γ to suppress tumor progression and angiogenesis [42,43,44]. In addition, this study found that the *FCGR3B* expression level accurately predicts the TB status (AUC = 81.7%) and patient survival (high expression causes poor prognosis in OS, *p* = 0.0016). Therefore, *FCGR3B* may be a novel biomarker for high TB in CC.

Second, low *ROBO2* expression may lead to a high-TB status through the axon guidance signaling pathway. The *ROBO2* gene encodes roundabout guidance receptor two protein that functions as an axon guidance receptor by binding to secreted SLIT ligands [45,46,47]. The ROBO family is frequently downregulated in several cancers and considered an anti-oncogene [48]. The SLIT/ROBO signaling can inhibit tumor progression through some mechanisms, such as preventing cell migration and angiogenesis, enhancing cell–cell adhesion, and blocking endothelial cell proliferation [49]. Interestingly, this signaling can regulate macrophage immune responses by inducing cytoskeletal changes in macrophages, preventing macrophage spreading and inhibiting macropinocytosis [50]. Although the *ROBO2* expression level in this study was not significantly related to patient survival prognosis because of the small sample size, *ROBO2* can still be a promising marker for predicting TB status (AUC = 92.2%).

Lastly, *NR4A2* (nuclear receptor subfamily 4 group A member 2) and *OPRL1* (opioid-related nociceptin receptor 1) were significantly enriched in several important biological processes, such as signal transduction, signaling receptor binding, signaling receptor activity, cell–cell signaling, and regulation of cell death. These findings suggest that *OPRL1* and *NR4A2* downregulation may enhance TB progression in CC. In other words, these genes function as tumor suppressors. With substantial diagnostic accuracy from ROC analysis, *OPRL1* and *NR4A2* are promising candidates for predicting TB status. Although there are several conflicting data regarding the role of NR4A2 and OPRL1 in cancer, our results are in accordance with current observations: NR4A2 can trans-activate Foxp3, involved in the differentiation, maintenance, and function of regulatory T cells, and plays a significant role in cancer cell development and survival [51,52,53]. Furthermore, Inamoto et al. reported that NR4A2 is a tumor suppressor in human bladder cancer tissues [51]. Regarding OPRL1, it belongs to the Aγ family of G protein-coupled receptors, which are involved in various diseases, including cancer [54,55]. Bedini et al. revealed that OPRL1 acts as a tumor inhibitor in U87 glioblastoma cells by blocking lipopolysaccharide [56]. In addition, OPRL1 can activate markers on the surface of T cells, enhance CD4+ T and CD8+ T-cell proliferation, and alter cytokine secretion, which is closely involved in tumor progression [57].

This study lacks experimental validation. Therefore, further studies should be conducted to clarify the mechanisms of these candidate genes in TB progression.

## 5. Conclusions

Four immune-related genes (one upregulated (*FCGR3B*) and three downregulated (*ROBO2*, *OPRL1*, and *NR4A2*) genes) that can impact TB formation and development were detected through a multiple-step analysis. These genes may be potential biomarkers for replacing TB status to stratify CC patients. Furthermore, they can provide novel ideas for immune-targeted therapeutic strategies in the future.

## Figures and Tables

**Figure 1 genes-13-01405-f001:**
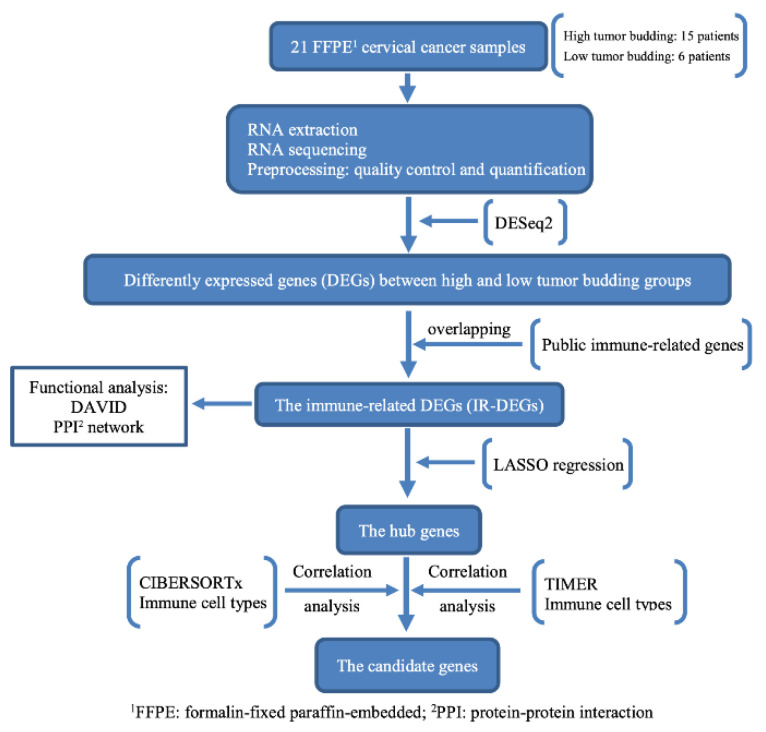
Flowchart of the study.

**Figure 2 genes-13-01405-f002:**
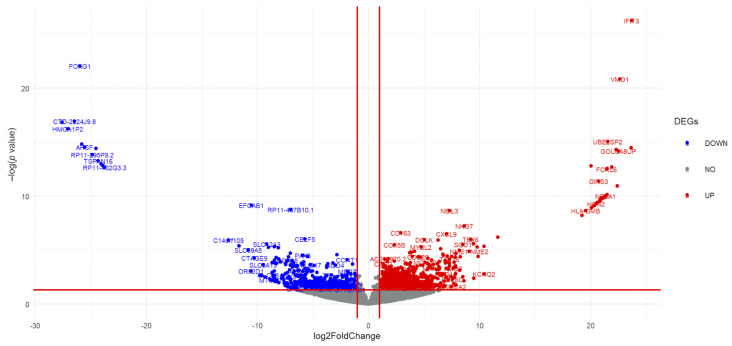
Differentially expressed genes (DEGs) between the high- and low-tumor-budding groups in cervical cancer. Blue and red represent downregulated and upregulated genes, respectively.

**Figure 3 genes-13-01405-f003:**
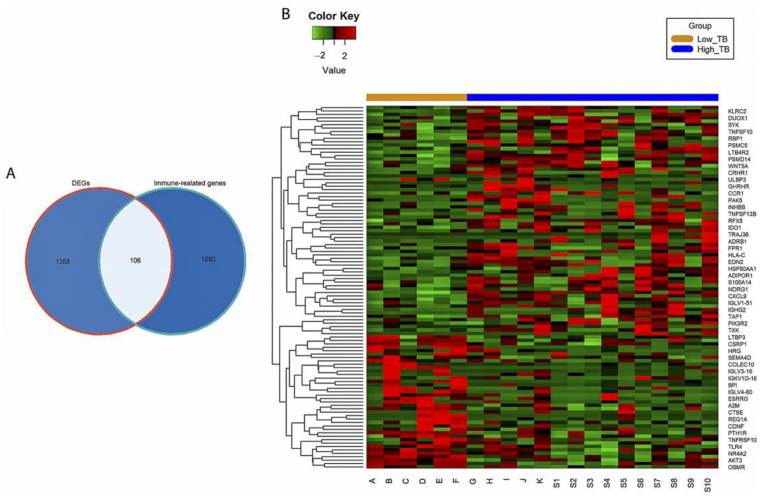
Immune-related differentially expressed genes (IR-DEGs). (**A**) Venn diagram of overlap genes between DEGs and public immune-related genes. (**B**) The heatmap of 106 IR-DEGs between high and low tumor budding (TB) in cervical cancer. Green, red, orange, and blue represent a low expression level, high expression level, low-TB group, and high-TB group, respectively.

**Figure 4 genes-13-01405-f004:**
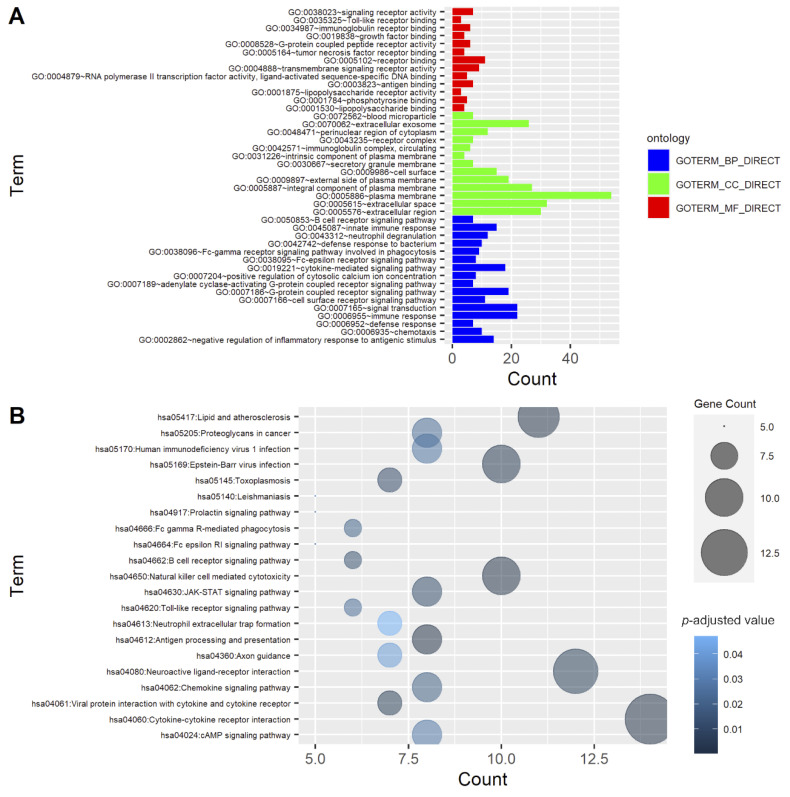
Functional enrichment analysis of 106 immune-related differentially expressed genes (IR-DEGs). (**A**) Gene Ontology (GO). The bar plot shows the top 50 enriched IR-DEGs from GO analysis. Blue, green, and red represent biological process (BP), cellular component (CC), and molecular function (MF) GO terms, respectively. (**B**) Kyoto Encyclopedia of Genes and Genomes (KEGG) pathway. Different colors and sizes of bubbles represent different *p* values and gene counts of a pathway.

**Figure 5 genes-13-01405-f005:**
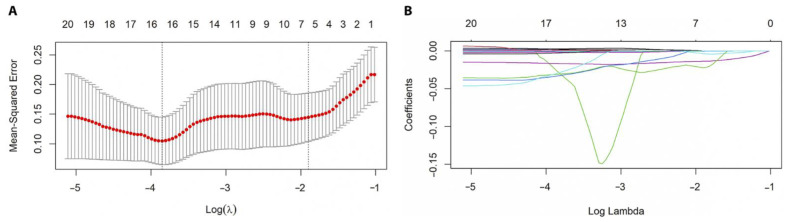
Least Absolute Shrinkage and Selection Operator (LASSO) regression analysis. (**A**) The minimum mean squared error was achieved and 16 hub genes were identified at the best lambda = 0.022. (**B**) When the log (λ) value increased, all coefficients shrunk precisely to zero.

**Figure 6 genes-13-01405-f006:**
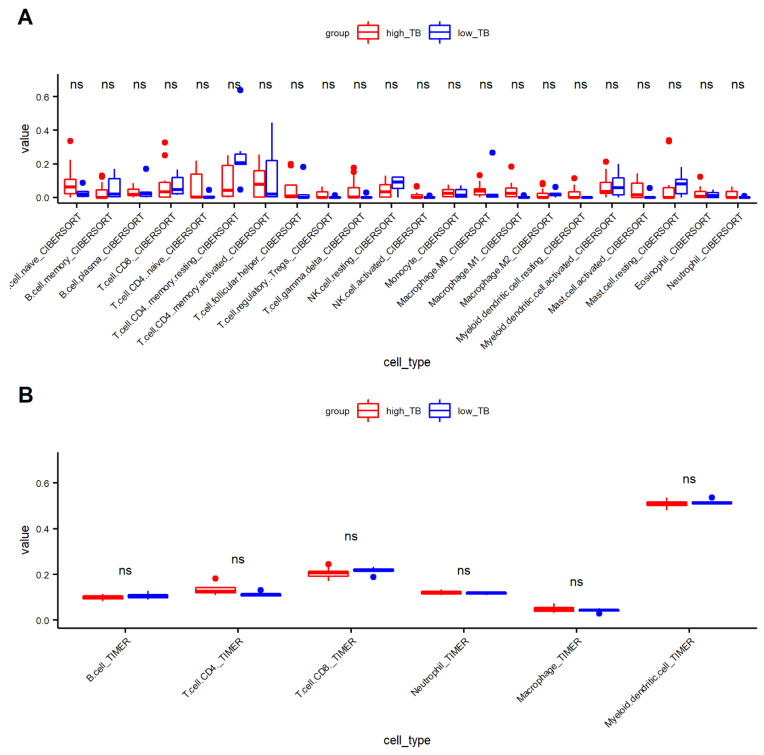
Distribution of immune cell types between high- and low-tumor budding (TB). (**A**) CIBERSORT and (**B**)TIMER. (NK.cell, natural killer cell; ns, non significant).

**Figure 7 genes-13-01405-f007:**
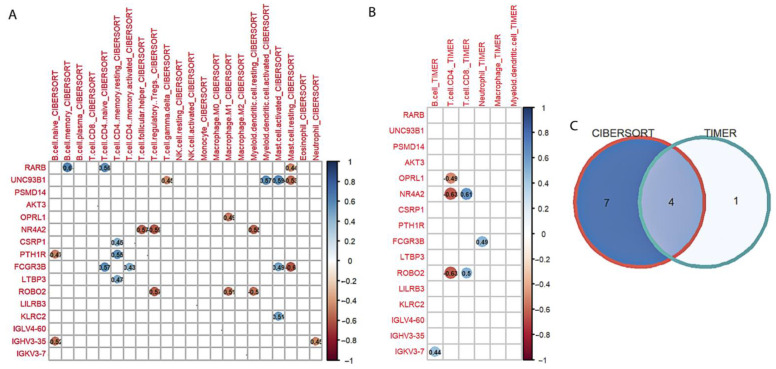
Identification of candidate genes. (**A**) Correlation between 16 hub genes and CIBERSORT immune cell types. (**B**) Correlation between 16 hub genes and TIMER immune cell types. (**C**) Venn diagram of overlap genes. Red and blue circles represent the significant negative and positive correlation, respectively. (NK.cell, natural killer cell).

**Figure 8 genes-13-01405-f008:**
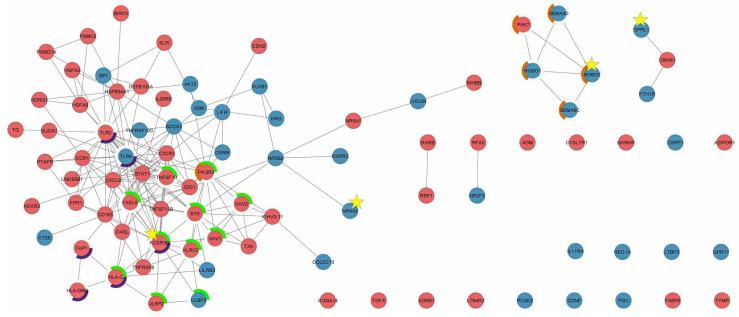
Protein–protein interaction network of 106 immune-related differentially expressed genes. “Red” and “blue” represent the upregulated and downregulated genes, respectively. The gray lines (edges) indicate interactions between connected nodes. Star highlights the candidate gene. The green, purple, and orange circles represent the natural-killer-cell-mediated cytotoxicity, phagosome, and axon guidance pathway, respectively.

**Figure 9 genes-13-01405-f009:**
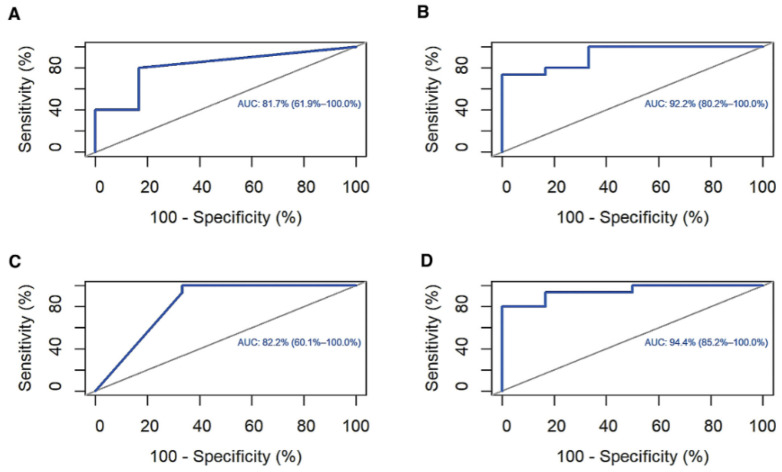
Receiver operating characteristic (ROC) curve analysis. (**A**) Fc region γ receptor III-B (FCGR3B), (**B**) roundabout guidance receptor 2 (ROBO2), (**C**) opioid-related nociceptin receptor 1 (OPRL1), and (**D**) nuclear receptor subfamily 4 group A member 2 (NR4A2).

**Figure 10 genes-13-01405-f010:**
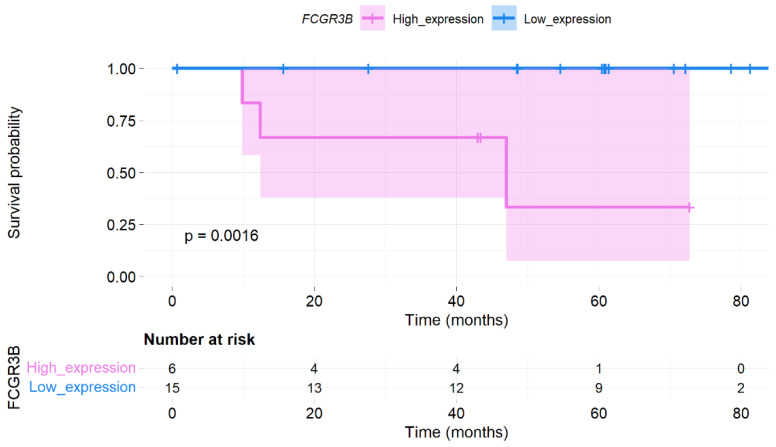
Survival analysis. Kaplan–Meier plot for Fc region γ receptor III-B (FCGR3B) expression level in overall survival.

**Table 1 genes-13-01405-t001:** Patients’ characteristics.

	(ALL)	N
	N = 21	
Age (range)	48 (32–71)	21
Clinical Stage:		21
Early stage	17 (81.0%)	
Late stage	4 (19.0%)	
Histology:		21
Squamous cell carcinoma	15 (71.4%)	
Adenocarcinoma	5 (23.8%)	
Adenosquamous cell carcinoma	1 (4.8%)	
Tumor budding:		21
Low (<5 TBs ^1^)	6 (28.6%)	
High (≥5 TBs)	15 (71.4%)	

^1^. TBs, tumor buds.

**Table 2 genes-13-01405-t002:** Kyoto Encyclopedia of Genes and Genomes pathways interacted with four candidate genes.

Term Name	Description	Genes	FDR
hsa04650	Natural killer cell mediated cytotoxicity	11	9.34 × 10^−9^
hsa04080	Neuroactive ligand-receptor interaction	12	4.44 × 10^−6^
hsa05152	Tuberculosis	8	5.28 × 10^−5^
hsa04380	Osteoclast differentiation	7	6.87 × 10^−5^
hsa04360	Axon guidance	7	4.20 × 10^−4^
hsa04145	Phagosome	6	9.10 × 10^−4^
hsa05150	Staphylococcus aureus infection	5	9.10 × 10^−4^

**Table 3 genes-13-01405-t003:** Top 10 Gene Ontology (GO) terms interacted with four candidate genes.

Category	Term Name	Description	Count	FDR
GO–BP ^1^	GO:0007165	Signal transduction	73	1.60 × 10^−^^23^
GO–BP	GO:0006955	Immune response	42	2.38 × 10^−^^18^
GO–MF ^2^	GO:0005102	Signaling receptor binding	41	2.81 × 10^−^^17^
GO–MF	GO:0038023	Signaling receptor activity	36	3.54 × 10^−^^14^
GO–BP	GO:0050776	Regulation of immune response	27	4.30 × 10^−^^12^
GO–BP	GO:0048584	Positive regulation of response to stimulus	40	8.52 × 10^−^^12^
GO–CC ^3^	GO:0009986	Cell surface	26	1.49 × 10^−^^11^
GO–BP	GO:0006935	Chemotaxis	20	5.35 × 10^−^^10^
GO–MF	GO:0008528	G protein-coupled peptide receptor activity	12	6.22 × 10^−^^09^
GO–BP	GO:0051239	Regulation of multicellular organismal process	42	2.67 × 10^−^^08^

^1^ GO–BP, Gene Ontology biological process; ^2^ GO-MF, Gene Ontology molecular function; ^3^ GO-CC, Gene Ontology cellular component.

## Data Availability

Not applicable.

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
