# Peer review of "Transcriptomic Immune Profiles Can Represent the Tumor Immune Microenvironment Related to the Tumor Budding Histology in Uterine Cervical Cancer"

_genes, 2022, doi:10.3390/genes13081405_

Round 1

Reviewer 1 Report

The focus of this review will be of substantial interest to researchers engaged in basic and clinical sciences. A more structured revised version with additional background information in a language accessible to non-specialist readers would substantially improve the value and impact of this article. Any revision of the manuscript should address the comments listed below. The manuscript is not well-written and needs careful copy editing.

Comments:

  1. Authors are required to modify and improve the abstract to make it more coherent to the findings.
  2. Give some background information about the tumor microenvironment and mention its components and importance for CC progression in the background section.
  3. Please add the latest references for every scientific statement mentioned in the article. For example, Line 251 should have a reference at the end.
  4. For a broader perspective, authors are required to correlate the four candidate genes with immune cell functions in other solid tumors if not known in CC. For instance, Robo/Slit signaling regulates macrophage polarization and its activity, which is crucial for tumor progression.
  5. Please mention in the methods how the heatmap was constructed.
  6. Authors are not required to give a full form for the abbreviation at all places. For example, DEG has been abbreviated more than 3 times in the manuscript.
  7. Please give the full form of Uterine cervical cancer (CC) in the abstract.
  8. The manuscript has many typographical and grammatical errors.

Author Response

Dear Reviewer,

We appreciate your time for the review and comments. We would like to submit our response to all your comments.

Thank you.

Sincerely,

Authors.

Reviewer 2 Report

Tan Minh Le et al., investigated the interactions between tumor budding (TB) and components of the tumor-immune microenvironment (TIME) in uterine cervical cancer tissues.

For this purpose, the authors collected 21 uterine cervical cancer tissues from the storage. High and low tuber budding were categorized by based on H&E staining images.

Total RNA extraction and sequencing were done from the samples. The results showed 1,494 DEGs between high and low TB groups.

Finally, the results confirmed up-regulation of FCGR3B and down-regulation of ROBO2, OPRL1&NR4A2 genes were correlated with tumor budding in uterine cervical cancer tissues.    

Please refer to the attached file for more comments.

Author Response

(The authors gave the same response as above.)
